# Health-Promoting and Sustainable Behavior in University Students in Germany: A Cross-Sectional Study

**DOI:** 10.3390/ijerph20075238

**Published:** 2023-03-23

**Authors:** Andrea Weber, Katharina Kroiss, Lydia Reismann, Petra Jansen, Gunther Hirschfelder, Anja M. Sedlmeier, Michael J. Stein, Patricia Bohmann, Michael F. Leitzmann, Carmen Jochem

**Affiliations:** 1Department of Epidemiology and Preventive Medicine, University of Regensburg, Franz-Josef-Strauß-Allee 11, 93053 Regensburg, Germany; 2Faculty of Human Sciences, University of Regensburg, Universitätsstraße 31, 93053 Regensburg, Germany; 3Faculty of Languages, Literature and Cultural Studies, University of Regensburg, Universitätsstraße 31, 93053 Regensburg, Germany

**Keywords:** health promotion, sustainability, active transport, sustainable diet, university setting, climate-specific health literacy, co-benefits, planetary health

## Abstract

Health-promoting and sustainable behaviors, such as active transportation and sustainable diets, are associated with positive effects on human health and the environment. In order to unlock the potential of university students as key actors and multipliers, it is of interest to investigate their level of knowledge about the health effects of climate change and their willingness toward and implementation of health-promoting and sustainable behaviors. In November 2021, an online survey was conducted among students at the University of Regensburg, Germany. A total of 3756 participants (response rate 18%; mean age 23 years; 69% women) provided valid data. A large proportion of medical students (48%) considered themselves well-informed about the health-related effects of climate change, while only a small proportion (22%) of students within economic/computer/data sciences and law felt informed. Most participants knew about the impact of climate change on malnutrition (78%), but considerably fewer were aware of its impact on cardiovascular diseases (52%). Participants who considered themselves informed were consistently more willing to engage in climate-friendly behavior, and this willingness was also reflected in their actions, as they simultaneously promoted a healthy lifestyle. Across all academic disciplines, there is a strong need for knowledge transfer regarding topics that combine health and sustainable development.

## 1. Introduction

Non-communicable diseases, such as cardiovascular disease, cancer, type 2 diabetes, and chronic respiratory disease, cause 74% of all deaths worldwide and lead to an enormous global burden of disease [1]. Lifestyle-associated, avoidable risk factors, such as an unhealthy diet and physical inactivity, are among the leading risk factors for non-communicable diseases [1]. In addition to their adverse health consequences, these unhealthy lifestyles have unfavorable effects on our natural environment and they foster man-made planetary crises, including climate change, losses in biodiversity, and pollution [2,3,4]. These environmental changes, in turn, have direct and indirect negative effects on human health [2,3,4]. Thus, human behavior and activities that are neither health-promoting nor sustainable threaten planetary health (defined as “the health of human civilisation and the state of the natural systems on which it depends” [5]) and impede the achievement of several of the United Nation’s Sustainable Development Goals (such as targets 3.4 and 12.8) [1,6]. Combining health-promoting activities with appropriate mitigation strategies leads to co-benefits for human health and environmental protection and may thereby boost planetary health [7]. Specifically, active transportation and sustainable diets are associated with positive effects on human health (e.g., by decreasing the risk of developing non-communicable diseases [8]) and on the environment (e.g., by decreasing levels of greenhouse gas emissions and pollution) and are among the key levers for achieving sustainable development [9,10,11].

Even if public awareness of these co-benefits is increasing, there is still a lack of broad social implementation of corresponding behavior [12]. However, a strong movement within broader society is necessary for a successful transformation [12], and certain populations (such as health professionals) play a key role [13]. University students can also become change agents, as they can act as multipliers through future professional and social positions. Furthermore, many young people identify with or belong to social movements that aim to contribute to sustainable development, such as the Fridays for Future movement. Although university students possess great potential to contribute to sustainable development and planetary health [14,15], this relatively large population subgroup has, thus far, hardly been studied in terms of knowledge/awareness of the health effects of climate change and other global environmental changes as well as the willingness to engage in health-promoting and sustainable behavior. Existing studies on planetary health literacy have focused primarily on students in health-related fields (e.g., medicine, nursing), and studies on sustainable development have not explicitly addressed health co-benefits.

In order to investigate knowledge and awareness of the health effects of climate change and willingness to implement health-promoting and sustainable/climate-friendly behavior, we conducted a cross-sectional study among university students across a wide range of academic areas of study in Germany. We aimed to address the following research questions:What is the level of knowledge about the health effects of climate change among students from different academic disciplines and how does this level of knowledge relate to their willingness to engage in climate-protective behavior?How is the theoretical willingness for sustainable behavior reflected in everyday actions and which health-promoting lifestyles are associated with such actions?

## 2. Materials and Methods

### 2.1. Setting and Time

Our questionnaire-based, cross-sectional study was conducted in November 2021 among students of the University of Regensburg (*n* = 20,678). The link for the online questionnaire and a reminder were sent to all students via the email distribution list of the University of Regensburg. The study protocol was approved by the Ethics Committee of the Medical Faculty of the University of Regensburg (project No. 21-2542-104, 4 August 2021). All participants provided their informed consent for the anonymous and voluntary data collection.

### 2.2. Study Instrument

Our German-language questionnaire was designed as part of a doctoral thesis on student health (URStudisHealthSurvey) to assess lifestyle, health, requirements, and health-specific climate literacy, and it included the following key aspects: (1) sociodemographic data; (2) dietary habits; (3) alcohol consumption, psychoactive substances and smoking; (4) physical activity and health promotion at the University of Regensburg; (5) physical and psychological impairments; (6) study and well-being; (7) global health and sustainable behavior; and (8) desires for the future of the university campus. The current manuscript focuses on the collected data on global health (part 7) in relation to health behaviors (parts 2 and 4) (see Appendix A).

For some questions (e.g., those related to smoking behavior, alcohol consumption, health promotion, and psychical and psychological impairments) the healthy campus questionnaire of the University of Bonn (Germany) and the German Sport University (Cologne) provided an orientation (https://www.uni-bonn.de/de/universitaet/ueber-die-uni/gesundheitsmanagement-healthy-campus (accessed on 20 January 2023)). The ten guidelines of the German Nutrition Society (DGE) for a wholesome diet (with a focus on diversified diets, fruits/vegetables, whole grain and dairy products, fish, vegetable oils, water, reduced meat, fat, salt and sugar intake, gentle food preparation, and mindful eating) were used as the basis for questions on healthy eating [16]. In addition, two items on buying regional, seasonal, or organic products and on cooking at home were added. For the assessment of physical activity and sedentary behavior, the Global Physical Activity Questionnaire (GPAQ) of the World Health Organization (WHO) provided guidance [17]. For consistency with current WHO recommendations, which emphasize that any physical activity is beneficial to health, no minimum duration was required [18]. Additionally, means of transport on the way to and from the university were collected. Questions on climate-specific health literacy (i.e., knowledge about the health consequences of climate change) and willingness to adopt climate-friendly behaviors in everyday life were taken from a previously published questionnaire by Reismann et al., which showed good face validity [19]. The questionnaire was implemented as an online form using LimeSurvey Pro. Categorical data were collected using single- or multiple-choice format or by using scales consisting of several Likert-type items. To assess continuous data, sliders with plausibility checks were mainly used.

### 2.3. Data Processing and Analysis

Body mass index (BMI) was classified according to WHO categories: <18.5 kg/m^2^ as “underweight”, 18.5–24.9 kg/m^2^ as “healthy weight”, 25.0–29.9 kg/m^2^ as “overweight”, and ≥30.0 kg/m^2^ as “obese” [20]. Compliance with the WHO recommendations for physical activity was established: according to the GPAQ analysis guide, physical activity of moderate intensity was assigned a MET value (metabolic equivalent of task) of 4, and physical activity of vigorous intensity was assigned a MET value of 8 [21]. The WHO’s recommendations were considered to be achieved if participants accumulated at least 600 MET minutes (equivalent to 150 min of moderate-intensity physical activity, 75 min of vigorous-intensity physical activity, or a combination of both) per week and reported muscle-strengthening activities at least twice a week [18]. A score for health-promoting nutrition was calculated by summing the fulfilled recommendations of the German Nutrition Society [16]. Accumulating 0 points indicated that no recommendations were complied with, and accumulating 13 points indicated that all recommendations were complied with. Continuous data were winsorized at the 99th percentile to reduce the impact of points of influence.

Data analysis used descriptive and exploratory methods. Categorical variables were reported as frequencies and percentages, and continuous variables were reported as means with standard deviation. Relationships between two or more variables are presented visually (e.g., by means of bar charts or alluvial plots). In analyses stratified by gender, students indicating diverse gender identities were excluded to protect participant anonymity. In all other analyses, participants of all genders were included. Regression models, adjusted for potential confounding variables, were constructed (logistic regression for binary responses; ordinal regression for Likert-type responses). A *p*-value of < 0.05 was considered statistically significant. Data processing and analysis were conducted using R Statistical Software version 4.2.1 [22].

## 3. Results

### 3.1. Study Population

Of 20,678 students enrolled at the University of Regensburg in the winter term 2021/2022, 4214 responded to the online questionnaire. After excluding 458 participants with missing or implausible data for student status, gender, or age, the analytic sample was comprised of 3756 participants (response rate 18%) (see Appendix A).

### 3.2. Participant Characteristics

Participant characteristics are shown in Table 1. The mean age was 22.7 (standard deviation (SD) = 4.3) years. With a proportion of 69%, women were most frequently included in the sample, which reflects the overall proportion of women at the University of Regensburg (60%). Further, the participation of students from different departments was, on the whole, representative of the respective student numbers. The largest proportion of study participants represented those in their 1st to 2nd study term (23.1%), and proportions decreased with an increasing number of terms (e.g., only 5.0% of study participants reported > 12 study terms). Regarding health behavior, the majority of participants were a healthy weight (74%) according to the WHO classification (mean BMI = 22.4 m/kg^2^ (SD = 3.5 m/kg^2^); *n* = 241 missing). Only 10% of participants reported smoking (*n* = 370 missing), but most participants reported drinking alcohol (75%, *n* = 345 missing). The 2020 WHO recommendations for physical activity were met by 40% of participants who answered this question (*n* = 551 missing). Students reported sitting for an average of 8.1 h/d (SD = 2.7 h/d; *n* = 551 missing), 48% of which, on average, was spent at the university. In a typical week, participants followed, on average, 5.2 (SD = 2.5; *n* = 309 missing) out of 13 rules based on the DGE guidelines for healthy eating.

### 3.3. Information about the Health Consequences of Climate Change

When asking participants which health consequences of climate change they had already heard about, malnutrition (78%), infectious diseases (73%), and respiratory complaints (67%) were the most frequently affirmed. Fewer participants had heard about the impact of climate change on cardiovascular problems (52%), heat shock/heat stress (57%), and increasing allergies (59%). In total, 31% of students indicated that they were well informed about these and other health consequences of climate change, whereas 10% of respondents were not aware that climate change can have a negative impact on health (*n* = 703 missing). Stratifying the level of information about the health effects of climate change by participants’ area of study showed that participants studying medicine, biology, chemistry, or pharmacy were the most aware; students of computer/data science, economic science, and law had heard the least about the various health impacts of climate change (Table 2). Data on informedness, stratified by age and gender, can be found in the Appendix A. On the whole, the proportion of informed people increased with age, and men felt slightly better informed than women.

Among 3196 students, the most interest was expressed for university courses on relaxation (63%), sports and exercise (60%), nutrition (49%), cooking classes (48%), health diagnostics (36%), and mindfulness (33%).

### 3.4. Willingness to Act in a Climate-Friendly Way

Most participants stated they were rather willing or very willing to reduce their ecological footprint (81%), use public transport (81%), use a bicycle or e-bike (78%), or eat vegetarian (75%). In contrast, only 44% of students were willing to mostly avoid animal products, e.g., via a vegan diet, in their daily lives (*n* = 704 missing). Participants with a higher level of knowledge about the health effects of climate change were generally more willing to implement various climate-friendly behaviors in everyday life (Figure 1). After adjusting for gender, age, and department, being well informed about the health consequences of climate change was statistically significantly associated with a willingness to reduce their ecological footprint, to use public transport or a(*n*) (e-)bike, and to adopt a vegan or vegetarian diet (*p*-value < 0.0001 for each; *n* = 3025).

### 3.5. Willingness versus Action: Transport

Contrasting the theoretical willingness to use a bicycle or e-bike with the actual means of transport on the way to university showed good conformity (Figure 2; *n* = 789 missing): Participants willing to use a bicycle/e-bike (*n* = 2323) mostly reported commuting to and from the university by foot or by bicycle/e-bike in the summer (65%). Participants who were not willing to use a bicycle/e-bike (*n* = 644) mainly commuted to and from the university using a motorized means of transport (summer 75%, winter 85%).

In both groups, there was a decrease in the prevalence of cycling in the winter compared to the summer. In the subgroup of participants willing to use a bicycle/e-bike, most summer cyclists also biked in the winter (53%) and 35% switched to public transport; in the subgroup of participants unwilling to use a bicycle/e-bike, the majority of summer cyclists switched to public transport (52%). On average, participants took 30 min (SD = 18 min) to get to the university and back by bicycle. Of the participants willing to use public transport, the majority commuted to and from the university by bicycle (38%) and public transport (36%) in the summer and by public transport (54%) in the winter (Appendix A). Students unwilling to use public transport reached the university predominantly by bicycle (32%) and car (27%) in the summer and mostly by public transport (34%) and car (33%) in the winter. Participants who reported walking or (e-)biking to the university accumulated more MET minutes per week than participants who traveled by public transport or car. For example, in the winter, bikers achieved, on average, 3976 (SD = 4033) MET minutes per week, while car drivers achieved, on average, 2717 (SD = 3639) MET minutes per week. Students using active means of transport were more likely to fulfill WHO recommendations for physical activity than participants reporting passive transport choices (*p* < 0.0001 after adjustment for age, gender, and department; *n* = 3074).

### 3.6. Willingness versus Action: Diet

Among 2298 participants willing to abstain from meat, a greater proportion of subjects (54%) reported consuming no more than 300 to 600 g of meat per week than among those 748 participants unwilling to adopt a vegetarian diet (29%). Students for whom a healthy diet was important also reported consuming less meat (50%; *n* = 2589) than students without this motivation (39%; *n* = 457). The largest group of students stated that a healthy diet was important to them, that they were willing to adopt a vegetarian diet, and that they consumed no more than 300 to 600 g of meat per week (*n* = 1121; *n* = 710 missing) (Appendix A). Further, students who reported a preference for shopping regional, organic, or seasonal products, on average, met more dietary recommendations (5.7 (SD = 2.4)) than students without these considerations (4.4 (SD = 2.3); *n* = 309 missing). Of the 2301 participants willing to abstain from meat, 61% stated that the university’s catering did not offer enough vegetarian or vegan options (*n* = 1193 missing). In total, 26% of students (*n* = 853 missing) stated there were not enough vegetarian/vegan options, and 27% (*n* = 729 missing) stated that there were not enough healthy offerings available on campus.

## 4. Discussion

Informedness of the health effects of climate change differed by field of study. Students from the medical or natural sciences reported being better informed compared to students from law or the economic or computer sciences. Furthermore, knowledge regarding specific health consequences, such as malnutrition or infectious diseases, was greater compared to knowledge regarding the link between climate change and cardiovascular diseases. With respect to willingness to act, most participants stated they were willing to reduce their ecological footprint by using public transportation, active transport, or employing a vegetarian diet. Students who reported being well-informed about the health effects of climate change indicated a greater willingness to adopt a less carbon-emissive lifestyle. This willingness to act in a climate-friendly way was also reflected in behavior; for example in diet or mode of transport, which in turn was associated with health co-benefits.

These findings support the results of other studies that investigated knowledge of planetary health/environment and health-related topics. For example, Klünder et al. (2022) assessed previous knowledge of planetary health among university students [23]. They found that participants studying medicine, epidemiology, or public health had more knowledge of planetary health than students from other health-related fields of study, such as pharmacy, veterinary medicine, and dentistry. However, that study mainly focused on students from health-related study areas [23]. Another study showed that the assessment and prioritization of sustainable development goals differed according to the field of study. While engineering or production management students ranked economic growth as the most important, environmental engineering students prioritized the problems of health, hunger, and well-being [24]. Regarding willingness to adopt climate-friendly lifestyles, the findings of the present study are in line with the results of an exploratory study by Reismann et al. (2021) among patients of general practitioners and gynecologists in Germany [19]. For example, participants were more ready to use a(*n*) (e-)bike, public transport, or adopt a vegetarian diet than to adhere to a vegan diet. However, in the study by Reismann et al., informedness (i.e., via climate-specific medical advice) was only associated with the willingness to adopt a vegetarian diet [19]. In accordance with our findings, a recent study found that university students biking or walking to university accumulated more transport physical activity, followed by public transport and car. Biking or walking to the university was sufficient to achieve physical activity recommendations [25]. A study of German university students identified high automobile traffic, time, effort, and weather conditions as barriers to transport-related cycling [26]. In our study, the prevalence of cycling to university also decreased in the winter. A survey of Portuguese university students showed that most participants were willing to reduce meat consumption, primarily for environmental considerations [27]. Another recent study found that university students in California and Michigan rated reducing meat consumption as a less effective strategy for mitigating climate change than, for example, recycling [28]. Even so, students who based their diet on sustainability aspects reported a lower frequency of red meat intake, which is in line with our results. Likewise, sustainability seemed a greater motivator for reduced meat consumption than health [28].

Our findings show that across all areas of study, there is a strong need among students for knowledge transfer regarding topics that combine health and sustainable development in terms of education for planetary health and/or education for sustainable development. However, in Germany (and most other countries) education for sustainable development is not comprehensively integrated into the curricula of many fields of study [29]. Although education for planetary health is increasingly implemented in health-related disciplines, it is not yet an integral part of corresponding curricula. In order to achieve Sustainable Development Goal 4.7, which aims to “ensure that all learners acquire the knowledge and skills needed to promote sustainable development, including among others through education for sustainable development and sustainable lifestyles, […]” by 2030 [30], there is a strong and urgent need to comprehensively provide both education for sustainable development and a university setting that allows students to act in a sustainable and health-promoting manner. This requires a whole-of-institution approach that not only breaks up disciplinary educational approaches and allows inter- and transdisciplinary education (with participatory co-design options for university students) but also fosters health-promoting and sustainable behavior in the university setting by making the healthy and sustainable choice the easier choice (with regard to mobility, nutrition, and other consumption patterns).

For example, active transport to university should be promoted by ensuring safe infrastructure, providing rental bicycles and e-bikes, and linking with sustainable public transport [31]. The Bicycle Friendly University program awards U.S. higher education facilities that provide physical infrastructure, education programs, encouragement, incentives, and equitable laws to promote biking [32]. Also, healthy, sustainable, and affordable food options should be a priority on campus to make them accessible for all students [33,34]. In our study, almost one-third of students stated that the food offered at the university does not provide enough healthy or vegetarian/vegan options. Universities can also contribute to imparting knowledge, skills and values for healthy and sustainable development. They should train future educators, experts, and decision makers in such a way that they become planetary health-literate individuals, i.e., that they are able “to make judgments and take decisions regarding planetary health, across societies and for health-promoting, sustainable, and transformative actions. Planetary health literate individuals and societies are enabled to sustain and promote their own health, population health, and the planet’s health” [35].

However, this goes beyond knowledge transfer and includes strengthening attitudes and values that appreciate the strong interconnectedness of human health and well-being with the state of the natural systems. Furthermore, changing behavior is challenging, and personal norms are essential: according to the theory of planned behavior, intentions, which are determined by personal attitudes, subjective norms, and perceived behavioral control, are relevant for changing behavior [36]. In the framework of the stage model of self-regulated behavior change [37], which developed from the model of action phases [38] and was applied in research on pro-environmental behavior, personal attitudes also play an essential role. However, another theory, the value belief norm model [39], emphasizes the factors of values (biospheric, altruistic, and egoistic), beliefs, and personal norms as relevant for pro-environmental behavior. In addition to this, the newly developed framework of the two-pathway of pro-environmental pathway suggests that next to a normative path, where personal and social norms are essential, a relational path is included based on connectedness to nature, empathy, and compassion [40]. Those theories suggest that it is not enough to enhance students’ cognitive knowledge. This is a necessary first step, as the study presented here has shown. Still, personal norms, attitudes, and internal transformative qualities, such as connectedness to nature, empathy, and compassion must be trained [41]. This is in line with a framework of individual, collective, and systems-level change provided by Wamsler et al. [42].

The authors also suppose that an increase in the transformative qualities of awareness, connection, and insight (besides purpose and agency), are relevant internal transformation qualities and that subjective well-being is an intermediary factor. Those factors lead to a change in relationships and connections, which results in pro-social and pro-environmental behavior and the regeneration of human and planetary well-being [42]. One way to increase inner transformation is to reach a state of mindfulness [43], which can be described as the ability of a person to be aware of the present moment non-judgmentally [44,45]. Due to different theoretical assumptions, practicing meditation [46] and being attentive to every moment can lead to dispositional mindfulness [47]. Thus, it might be promising to establish university mindfulness courses to foster individual and planetary health consciousness subsequent to knowledge acquisition in education for sustainable development. Since one-third of students in our survey expressed interest in mindfulness, compliance is expected to be high.

In addition, it is important to consider sociocultural aspects that determine the knowledge-behavior gap: Although sustainability and animal welfare are currently at the center of the public discourse, the surveys reveal divergences between knowledge about climate-friendly nutrition and lifestyles and actual consumption decisions, for example in terms of veganism. This is also due to the fact that, beyond the material level, food is symbolically charged, traditionally shaped, and emotionally guided. On the other hand, a subjective definition of “healthy nutrition” proves to be problematic, since this is imagined differently in a multi-ethnic and multi-religious society, in which the social gap is also widening. For instance, migrants from the Arab region tend to have a greater affinity for sugar, which has a positive connotation, while migrants from Eastern Europe show a significantly higher affinity for meat. Instead of achieving profound changes through nudging and thus disregarding the diversity of culturally determined preferences, awareness should be raised that change toward more sustainable dietary practices is rather intrinsic and embedded in sociocultural transformations. Consequently, qualitative studies in which the social and ethnic backgrounds of the study participants, the associated divergent concepts of a diet and lifestyle that is considered healthy, and the symbolic character of food, the acquisition and consumption of which is also a result of processes of reflection and communication, should be expressed as major desiderata [48,49].

To the best of our knowledge, this is the first study to assess the theoretical knowledge, willingness, and implementation of sustainable and health-promoting behaviors among university students. A major strength is the large number of participants from a variety of academic disciplines, thereby enhancing the generalizability of our findings to similar populations, and the broad but detailed assessment of different aspects of planetary health. Nonetheless, our study has some limitations. Our results should be interpreted with caution because of the non-probability sampling method employed. Self-reported data could be affected by recall issues or social desirability reporting, which might lead to over- or underestimation of estimates. At the time of the survey, university lectures were held in attendance, however, it cannot be ruled out that results were influenced by the COVID-19 pandemic, e.g., by reducing the willingness to use public transport or contributing to unhealthy lifestyles.

## 5. Conclusions

The key finding of our study is that across all areas of study among university students, there is a strong need for knowledge transfer regarding topics that combine health and sustainable development. Besides knowledge transfer, university students should be enabled to develop and strengthen skills and values for healthy and sustainable development and to become planetary health literate individuals. With respect to nutrition and transport, a whole-of-institution approach is necessary to make health-promoting and sustainable patterns of nutrition and mobility the easier choices in the university setting. While our study provides initial exploratory results, further multicenter and multinational studies employing mixed methods approaches are needed.

## Figures and Tables

**Figure 1 ijerph-20-05238-f001:**
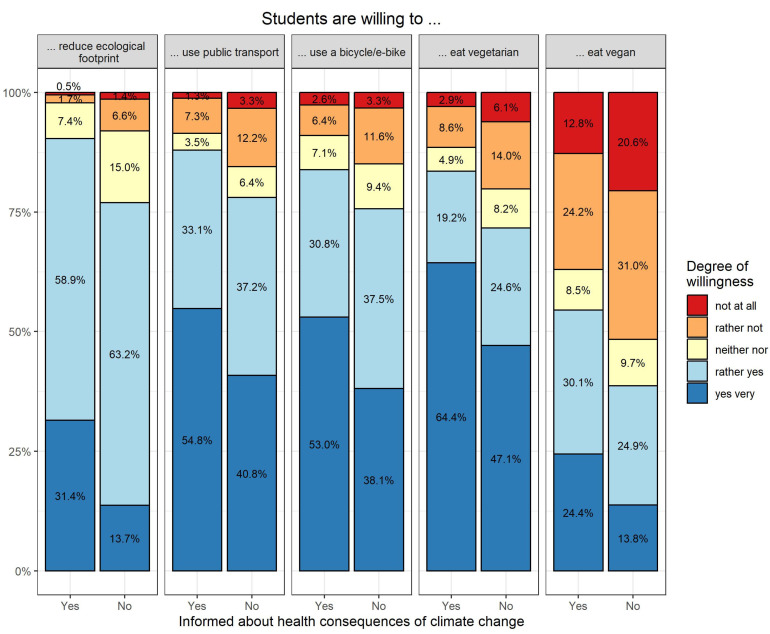
Willingness to engage in sustainable behaviors, stratified by informedness about the health consequences of climate change (*n* = 3052).

**Figure 2 ijerph-20-05238-f002:**
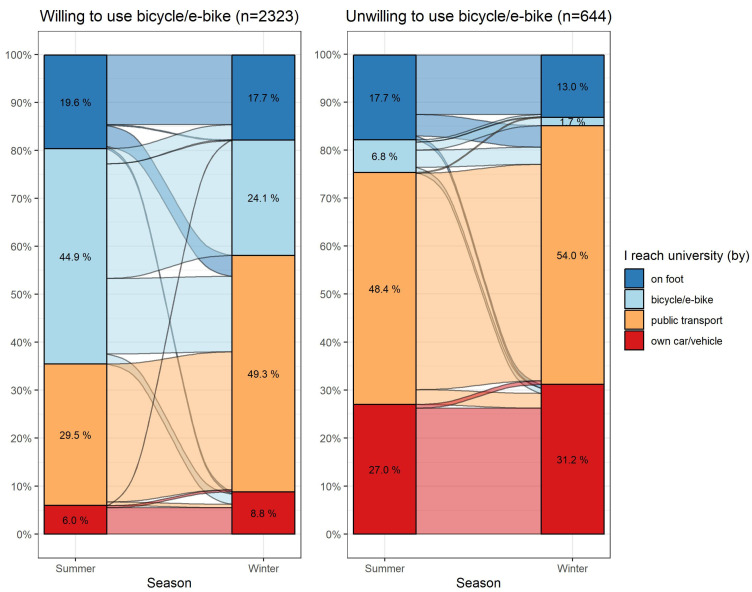
Means of transport commuting to and from the university in the summer and the winter, stratified by willingness to use a bicycle/e-bike (*n* = 2967). The curves between the bar charts represent changes in the composition of the strata between summer and winter.

**Table 1 ijerph-20-05238-t001:** Baseline characteristics of the study participants (*n* = 3756).

Variable (*n* Missing)	Levels	Mean (SD) or *n*
	Men	1140
Gender (0)	Women	2584
	Diverse	32
	<22 years	1686
Age category (0)	22–25 years	1534
	>25 years	536
Age in years (0)		22.7 (4.3)
	Catholic theology	79
	Law	334
	Economic sciences	316
	Medical studies	309
	Philosophy/arts/history/social studies	413
	Computer/data science	42
Department (0)	Human sciences	646
	Linguistics/literature/cultural studies	660
	Mathematics	159
	Physics	161
	Biology/preclinical medicine	376
	Chemistry/pharmacy	261
	1st to 2nd term	868
Study term (0)	3rd to 6th term	1469
	>6th term	1419
	Underweight	299
BMI category (241)	Healthy weight	2584
	Overweight	488
	Obesity	144
BMI in kg/m^2^ (241)		22.4 (3.5)
Drinking alcohol (345)	Yes	2542
	No	869
Smoking (370)	Yes	355
	No	3031
Physical activity MET minutes/week (551)		3252.0 (4123.8)
Muscle-strengthening activity times/week (551)		1.6 (1.7)
Comply with WHO physical activity recommendations (551)	Yes	1273
Sedentary behavior h/d (551)		8.1 (2.7)
	Diversified diet	2220
	Fruit/vegetables	765
Comply with DGE healthy	Whole grain products	1640
diet (309)	Dairy products	1566
	Fish	753
	Meat	1634
	Vegetable oils	1584
	Hidden fats	901
	Sugar	1723
	Salt	730
	Water	2496
	Gentle preparation	790
	Mindful eating	1016
DGE score [0; 13] (309)		5.2 (2.5)
Buying regional/seasonal/organic products (309)	Yes	1996
Cooking at home (309)	Yes	2667

*n*, number; SD, standard deviation; BMI, body mass index; MET, metabolic equivalent of the task; WHO, World Health Organization; DGE, German Nutrition Society.

**Table 2 ijerph-20-05238-t002:** Informedness by department: What health consequences of climate change have you already heard about? (proportion of affirmative answers; *n* = 3053).

Area of Study	Heatstroke/Heat Stress	Cardiovascular Issues	Increased Allergies	Mental Health Issues	Respiratory Symptoms	Global Malnutrition	Infectious Diseases	Well Informed about All of These	Not Aware of the Health Impacts of Climate Change
Catholic theology (*n* = 58)	40	47	55	48	55	66	67	28	19
Law (*n* = 268)	46	41	55	49	58	70	66	22	14
Economic sciences (*n* = 239)	50	41	46	49	57	64	59	22	18
Medical studies (*n* = 269)	70	67	74	76	81	88	90	48	2
Philosophy/arts/history/social studies (*n* = 334)	62	55	58	60	70	80	75	33	9
Computer/data science (*n* = 32)	38	31	41	50	47	59	41	22	31
Human sciences (*n* = 530)	50	48	59	64	66	79	73	25	9
Linguistics/literature/cultural studies (*n* = 517)	57	52	58	63	67	76	70	32	12
Mathematics (*n* = 137)	53	47	53	63	64	77	67	24	10
Physics (*n* = 134)	66	58	55	55	71	84	66	34	7
Biology/preclinical medicine (*n* = 322)	67	58	65	69	74	87	83	39	5
Chemistry/pharmacy (*n* = 213)	60	51	60	59	71	78	76	35	9
Total (*n* = 3053)	57	52	59	61	67	78	73	31	10

## Data Availability

The data presented in this study are available upon request from the corresponding author. The data are not publicly available due to privacy restrictions.

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
