# Peer review of "Health-Promoting and Sustainable Behavior in University Students in Germany: A Cross-Sectional Study"

_ijerph, 2023, doi:10.3390/ijerph20075238_

Round 1
Reviewer 1 Report
The authors state that the article investigates students' level of knowledge about the health effects of climate change and their willingness for and implementation of health-promoting and sustainable behaviors. For this, an online survey was conducted among students at the University of Regensburg, Germany (n 3756). The results show a strong need for knowledge transfer across all academic disciplines regarding topics that combine health and sustainable development.
The article is well-written and presents a structure that meets the proposed objective. I have a few suggestions:
1 - Why did the research include only students from one university? Justify.
2 - How can the study results benefit other schools or countries? Can the results be generalized?
3- The research was carried out amid the COVID-19 pandemic. Therefore, the answers may be biased due to the health crisis experienced during the period. Justify the period in which the study was carried out.
4 - Was the sampling probabilistic? Justify.
5 - In the conclusions, highlight the main contributions of the study, as well as suggestions for future research.
Reviewer 2 Report
The authors provided well-designed, thought-out research that involved University students and their perception of the health effects of climate change. The researchers delivered a substantial review of the literature to set the stage for the study's specific aims. The aims of the study were novel and pertinent to the literature.
Overall, the study involved an adequate sample size with a robust quantitative data analysis and results section. Although I felt figure one was well-displayed, I found figure two somewhat confusing. I am not sure how to decipher the curved lines. I recommend looking at a different way to display the results.
The discussion and conclusions were warranted and added to the literature. The researchers provided strong examples of literature to support their findings.
The references were up to date and relevant.
My one suggestion would be to conduct mixed methods approach in the future to provide additional qualitative data to support the quantitative data. That is just a suggestion for future research and not a reason to deny this paper.
Reviewer 3 Report
Overall description of manuscript
This manuscript reports a cross-sectional study through online survey on Germany college students (1 university) regarding their knowledge about health effects of climate change and their willingness for and their implementation of health-promoting as well as sustainable behaviors. The results show that majority of these college students were aware of the impact of global climate change on malnutrition, but only around half of the students realized that climate changes also affect on CVD risk. The knowledge about climate change-related issues of college students is positively related to their willingness to engage, and their actual actions and behaviors in implementing SDG and healthy lifestyle promotion.
Comments and suggestions to authors and editors
1. This manuscript is well written and enjoyable to read. Authors clearly point out the importance of evaluating “college students” for their knowledge, attitude and behavior related to combating climate change and implementing healthy lifestyle (Lines 49-62). The novelty of this research is considering college students as the key actors and multipliers for combating climate changes and health-promotion through implementing SDG and managing a healthy lifestyle since young age.
2. Authors also state their research questions clearly to be verified by the findings of the results.
3. Comparisons of results among students from a wide range of professional disciplines not only point out the directions for future education on health-promotion and building sustainable behavior for students, but also allow faculties in higher education a chance of reflection on future curriculum development for meeting the urgent needs of facing global climate changes and SDG implementation in each profession.
4. Authors report many critical issues which are very important for the applications of the results of this research in the 3rd paragraph of Discussion (page 9, starting from Line 280). However, the 3rd paragraph is so long as almost 1.5 pages. These issues include the importance of education, supporting environment, example such as bike friendly campus (Lines 280-301), enhancing the planetary health literacy (Lines 302-3100, related behavioral change theories, and value belief norm model and their applications (Lines 311-333), power of inner transformation and students’ interest in mindfulness, SDG and animal welfare (Line 343), cultural influence, … etc. I suggest that authors revising 3rd paragraph into many shorter paragraphs based on each critical issue discussed here. Therefore, each critical issue would stand out more obviously for itself.
Reviewer 4 Report
This is a well written manuscript exploring health-promoting and sustainable behaviour in university students in Germany.
I have only a few suggestions.
Results
3756 participants (18% students) – not sure about this one??? Only 18% of 3756 participants were students??? If it is response rate, then specify 18% response rate
Table 1 – There is no discussion about the education qualification/status i.e. undergraduate, post graduate of student participants
Also, I am curious to know whether all the participants were German nationals or there were some international students also??
Nutrition is part of which academic discipline?? Medical studies or Human sciences; Is sports science a part of the University curriculum?
Line 309 […] – Complete the sentence
